# Factors associated with COVID-19 among hospitalized patients with severe acute respiratory infections in Serbia, 2022–2023: A test negative case-control study

**Maja Stosic**[1]*, **Dragana Plavsa**[1], **Verica Jovanovic**[1], **Marko Veljkovic**[1], **Dragan Babic**[1], **Aleksandra Knezevic**[2], **Vladan Saponjic**[1], **Dragana Dimitrijevic**[1], **Miljan Rancic**[3], **Marija Milic**[1,4], **Tatjana Adzic-Vukicevic**[5,6]

1 Institute of Public Health of Serbia „Dr Milan Jovanovic Batut", Belgrade, Serbia, 2 Institute for Microbiology and Immunology, Faculty of Medicine, University of Belgrade, Belgrade, Serbia, 3 World Health Organization, Country Office Serbia, Belgrade, Serbia, 4 Department of Epidemiology, Faculty of Medicine, University of Pristina Temporarily Seated in Kosovska Mitrovica, Kosovska Mitrovica, Serbia, 5 COVID Hospital "Batajnica", University Clinical Centre of Serbia, Belgrade, Serbia, 6 Clinic for Pulmonology, University Clinical Centre of Serbia, Belgrade, Serbia

* maja_stosic@batut.org.rs

**Data Availability Statement:** The data underlying the results presented in the study are available and uploaded as a Supporting Information file.

## Abstract

Severe acute respiratory infections (SARI) are estimated to be the cause of death in about 19% of all children younger than 5 years globally. The outbreak of coronaviral disease (COVID-19) caused by SARS-CoV-2, increased considerably the burden of SARI worldwide. We used data from a vaccine effectiveness study to identify the factors associated with SARS CoV-2 infection among hospitalized SARI patients. We recruited SARI patients at 3 hospitals in Serbia from 7 April 2022–1 May 2023. We collected demographic and clinical data from patients using a structured questionnaire, and all SARI patients were tested for SARS-CoV-2 by RT-PCR. We conducted an unmatched test negative case-control study. SARS-CoV-2 infected SARI patients were considered cases, while SARS CoV-2 negative SARI patients were controls. We conducted bivariate and multivariable logistic regression analysis in order to identify variables associated with SARS-CoV-2 infection. We included 110 SARI patients: 74 were cases and 36 controls. We identified 5 factors associated with SARS-CoV-2 positivity, age (OR = 1.04; 95% CI = 1.01–1.07), having received primary COVID-19 vaccine series (OR = 0.28; 95% CI = 0.09–0.88), current smoking (OR = 8.64; 95% CI = 2.43–30.72), previous SARS CoV-2 infection (OR = 3.48; 95% CI = 1.50–8.11) and number of days before seeking medical help (OR = 0.81; 95% CI = 0.64–1.02). In Serbia during a period of Omicron circulation, we found that older age, unvaccinated, hospitalized SARI patients, previously infected with SARS CoV-2 virus and those who smoked, were more likely to be SARS-CoV-2-positive; these patient populations should be prioritized for COVID vaccination.

**Funding:** The author(s) received no specific funding for this work.

**Competing interests:** The authors have declared that no competing interests exist.

## Introduction

Acute respiratory infections (ARI) cause substantial morbidity and mortality every year, accounting for 6% of global disease burden [1, 2]. Since 2020, the coronavirus disease (COVID-19) pandemic has caused nearly 800 million reported infections and 7 million reported deaths worldwide [3].

Following the influenza, A (H1N1) pdm09 pandemic in 2009, the World Health Organization (WHO) encouraged countries to implement sentinel surveillance for severe acute respiratory infections (SARI) as a way to monitor the epidemiology and impact of influenza, and characterize risk factors for severe disease severe illness [4]. Following the start of the COVID-19 pandemic, WHO recommended that countries with existing sentinel SARI surveillance for influenza to expand these systems to monitor for COVID-19, in part to be able to estimate COVID vaccine effectiveness [5]. In the Republic of Serbia, SARI surveillance was established in 2009. In 2022, two SARI surveillance sites were expanded to include case detection for SARS-CoV-2 as part of study to estimate COVID-19 vaccine effectiveness [5–7].

Understanding difference in demographics, comorbidity profiles, and clinical course between hospitalized COVID patients compared to non-COVID patients can help inform preventive measures and clinical care. Previous studies have shown that hospitalized COVID patients differ from those with non-COVID diagnoses by age, chest X ray abnormalities, laboratory findings and lung function parameters [8–10]. In addition, studies have shown that compared to non-COVID patients, hospitalized COVID patients were more likely to have immunosuppression and cardiovascular disease and asthma, while asthma was associated with better clinical outcomes [11, 12]. SARS CoV-2 positive SARI patients have also been found to have a three-fold higher risk of mortality [4–12].

We used data from three enhanced SARI sentinel sites in Serbia to evaluate factors associated with SARS-CoV-2 infection among hospitalized SARI patients and to compare the clinical course of hospitalized SARS-CoV-2-positive SARI patients with SARS-CoV-2-negative patients. Two hospitals where the study was performed are part of the national SARI sentinel surveillance system in Serbia. Overall, six hospitals, (four treating adult patients and two child and adolescent ones) represent the national sentinel SARI surveillance network in Serbia. During the COVID-19 pandemic, based on the clinical protocol, patients with the most severe clinical SARI presentations from all over the country were referred and treated in these two hospitals included in the study. In addition, third sentinel site was the "Batajnica" Hospital, dedicated for treatment of the most severe COVID-19 patients during study period.

## Methods

### Study design

We performed an unmatched test negative case-control study using data from patients hospitalized with SARI at three hospitals in Serbia from 7 April 2022 to 1 May 2023. The hospitals included two SARI sentinel sites—The Clinic for Pulmonology and Clinic for Infectious and Tropical Diseases. Both are part of the University Clinical Center of Serbia in the capital, Belgrade. The third site was the "Batajnica" Hospital, which served as a referral hospital for severe COVID-19 patients during this period.

### Sample and procedure

We defined cases as persons at least 18 years old who were hospitalized, met WHO's SARI case definition [13] and had laboratory-confirmed SARS-CoV-2 infection for the first time either at the time of their hospital admission or within 14 days prior to the current hospital admission.

Controls were hospitalized persons who met the SARI case definition, tested negative for SARS-CoV-2 infection at the time of current hospital admission and had not tested positive for SARS-CoV-2 within 14 days prior to their current hospital admission. We defined hospital admission as a being in-hospital for a minimum of 24 hours.

We included all eligible consecutive patients fulfilling the SARI case definition, we accessed during the study period. Due to extreme workloads, it was only feasible to switch from exhaustive to systematic sampling (e.g. inclusion of patients only once a week, on certain days). Based on the national surveillance data, during the COVID-19 pandemic, frequency of severe acute respiratory infections was much higher among SARS CoV-2 positive patients then in SARS CoV-2 negative patients, more than five times and therefore the number of cases is higher than the number of controls.

## Inclusion and exclusion criteria

We included patients who met the SARI case definition and had the mental capacity to sign an informed consent form, or had a legally authorized representative who could sign the consent form. We excluded patients who could not be swabbed due to severe septum deviation, obstruction or other conditions that contra-indicated swabbing. We also excluded patients who refused to participate or did not have the mental capability to provide informed consent or a legally authorized representative who could provide consent on their behalf.

## Data collection

Health workers at each of the three hospitals identified SARI patients admitted to the hospital and administered a questionnaire to all SARI patients who provided consent. The survey included 70 closed-ended questions related to the following categories: socio-demographics; health behaviour; COVID-19, influenza and pneumococcal vaccination status; comorbidities; and symptoms of present illness. Health workers later collected data related to the clinical course of the current SARI-related hospitalization. Data were de-identified and stored in the REDCap data management application [14].

## Laboratory methods

All SARI patients had nasal or nasopharyngeal swabs collected for SARS-CoV-2 testing by reverse transcription-polymerase chain reaction (RT-PCR) within the first 48 hours of hospital admission, if they had not been tested by RT-PCR in 14 days before admission or if they had negative antigen rapid test. Testing was performed according to standard biosafety and biosecurity standards [15]. Samples with RT-PCR Cyclic threshold (Ct) values < 30 were sent to the Virology laboratory of the Institute of Microbiology and Immunology, Faculty of Medicine, University of Belgrade, where whole-genome sequencing (WGS) was performed to identify SARS-CoV-2 variants.

WGS of the selected samples was performed according to the ARTIC nCoV-2019 sequencing protocol v3 (LoCost) V.3 using Oxford Nanopore sequencing platform [16]. The prepared libraries were quantitatively checked, barcoded, and sequenced on a MinION sequencer, using an R9.4.1 flow cells (Oxford Nanopore Technologies, UK). The analysis of the MinION raw data was carried out according to the ARTIC nCoV bioinformatics SOP v.1.1.0 (https://artic. network/ncov-2019/ncov2019-bioinformatics-sop.html). The consensus sequences of SARS-CoV-2 were obtained using the assembly method Medaka v. v. 1.0.1; ARTIC nCoV-2019 v. V3 with ≈200x coverage. Derived genomes with related information were deposited in the Global Initiative on Sharing All Influenza Data (GISAID; https://www.gisaid.org/epiflu-ap-plications/ next-hcov-19-app/).

## Statistical analysis

We described baseline characteristics of cases and controls using mean and standard deviation (SD) for continuous variables and counts and proportions for categorical variables. We performed bivariate analysis to identify the crude association between dependent and independent variables. The dependent variable was the presence of SARS-CoV-2 infection and the independent variables included socio-demographic data, symptoms of present illness, vaccination, comorbidities and the clinical course of the current SARI-related hospitalization.

Statistical significance was determined using $p < 0.05$ as a cut-off point, and odds ratio was used as a measure of the strength of association. Variables which showed significant association (at p value $\leq 0.05$) in bivariate analysis, which were conceptually related to the outcome and preceded it, and were not in multicollinearity with the other variables, were entered in a logistic regression procedure for multivariable logistic analyses, in order to assess the independent predictors of SARS-CoV-2 infection among the study participants. We analysed data using IBM SPSS Software V20.0.

## Ethical approval

This study was approved by the Ethical Board of the Institute of Public Health of Serbia (No 6501/1) and the WHO Research Ethics Review Committee (CERC.0098D). All patients signed a written informed consent, and the research team signed the confidentiality statement [17].

## Results

During the study period, we enrolled 110 SARI patients, of whom 74 patients tested positive for SARS-CoV-2 (cases), and 36 tested negative (controls). Most patients (61.8%) were from COVID Hospital "Batajnica". The mean age was 71.6 ±15.6 for cases and 63.1±16.9 for controls. Males constituted 55.4% of cases and 63.9% of controls.

In the bivariate analysis, there were no significant differences between cases and controls with regards to sex, body mass index (BMI), education degree, occupation, marital status, residence and monthly income of the family. However, compared to controls, cases were significantly more likely to be older (p = 0.013), unemployed (Odds ratio (OR) = 2.02; 95% Confidence Interval (CI) = 1.26–3.24), current smokers (OR = 4.02; 95% CI = 1.50–10.82), former smokers (OR = 4.20; 95% CI = 1.73–10.19) and alcohol users (OR = 3.38; 95% CI = 1.47–7.77). The variable related to three or four comorbidities was very close to statistical significance (Table 1).

In bivariate analysis, fewer cases had received primary vaccine series of any type and any type of third dose compared to controls; however, the percentage of SARI patients who had received a booster dose was almost similar between cases and controls. The majority of study participants (37.8% among cases and 55.5% among controls) received Sinopharm BBIBP-CorV as the primary series (Table 2).

Controls took a median of one-half day more to seek medical help compared to cases (p = 0.044). Period from symptom onset till hospital admission was more likely shorter among cases (p <0.001). In addition, oxygen saturation on admission (p = 0.013) was higher among cases than controls as well as lowest recorded oxygen saturation during hospitalization (p = 0.006). More cases had previous laboratory-confirmed SARS-CoV-2 infection compared to controls (OR = 14.60; 95% CI = 5.37–39.70)—Table 3.

There was no difference in oxygen application method, frequency of individual co-morbidities and treatment outcome.

In multivariable logistic regression analysis, five variables were found to be significantly independent factors associated with COVID-19: age (OR = 1.04; 95% CI = 1.01–1.07),

**Table 1. Bivariate analysis of socio-demographic characteristics of cases and controls.**

| Variables | Cases (N = 74) | Controls (N = 36) | OR (95% CI) | p value* |
|---|---|---|---|---|
| | No (%) | No (%) | | |
| Hospital | | | | |
| COVID Hospital "Batajnica | 68 (91.9) | 0 (0.0) | na | na |
| Clinic for Pulmonology | 2 (2.7) | 23 (63.8) | | |
| Clinic for Infectious Diseases | 4 (5.4) | 13(36.2) | | |
| Sex | | | | |
| Male | 41 (55.4) | 23 (63.9) | ref | 0.398 |
| Female | 33 (44.6) | 13 (26.1) | 1.42 (0.63–3.23) | |
| Age (years) mean, SD | 71.6±15.6 | 63.1±16.9 | 1.03 (1.01–1.06) | 0.013 |
| ≤50 | 12 (16.2) | 7 (19.4) | ref | |
| >50 | 62 (83.8) | 29 (80.6) | 1.25 (0.45–3.50) | 0.675 |
| BMI (mean, SD) | 24.8 ±4.2 | 25.8 ± 6.3 | 0.96 (0.86–1.06) | 0.392 |
| Underweight | 7 (6.9) | 1 (5.6) | | |
| Normal | 37 (51.4) | 8 (44.4) | | |
| Overweight | 22 (30.6) | 6 (33.3) | | |
| Obesity | 8 (11.1) | 3 (16.7) | 0.79 (0.41–1.50) | 0.466 |
| Co-morbidities | | | | |
| Hypertension (HTA) | 50 (67.6) | 20 (55.6) | 1.67 (0.74–3.78) | 0.221 |
| Chronic cardiac disease, except hypertension | 22 (29.7) | 5 (13.9) | 2.57 (0.88–7.48) | 0.082 |
| Diabetes | 17 (23.0) | 9 (25.0) | 0.89 (0.35–2.26) | 0.814 |
| Cancer | 9 (12.2) | 4 (11.1) | 1.08 (0.32–3.87) | 0.873 |
| Kidney | 6 (8.1) | 2 (5.6) | 1.50 (0.29–7.83) | 0.631 |
| COPD and asthma | 14 (18.9) | 5 (13.9) | 1.45 (0.48–4.39) | 0.514 |
| One or more co-morbidities** | 61 (82.4) | 31 (86.1) | 0.87 (0.50–1.52) | 0.625 |
| Two co-morbidities | 25 (33.8) | 10 (27.8) | 1.15 (0.74–1.78) | 0.526 |
| Three and four co-morbidities | 14 (18,9) | 2 (5.6) | 1.99 (0.92–4.30) | 0.080 |
| Functional physical impairment before current illness? | 3 (4.1) | 2 (5.6) | 0.72 (0.12–4.50) | 0.724 |
| Current smoking | 33 (44.6) | 6 (16.6) | 4.02 (1.50–10.82) | 0.006 |
| Former smoking | 61 (82.4) | 19 (52.7) | 4.20 (1.73–10.19) | 0.002 |
| Alcohol consumption | 54 (73.0) | 16 (44.4) | 3.38 (1.47–7.77) | 0.004 |
| Education | | | | |
| High school or lower | 44 (59.5) | 20 (55.6) | ref | |
| University or higher | 30 (40.5) | 16 (44.6) | 0.85 (0.39–1.91) | 0.697 |
| Occupation | | | | |
| Employed | 10 (13.5) | 14 (38.9) | ref | |
| Unemployed/Retired | 64 (86.5) | 22 (61.1) | 2.02 (1.26–3.24) | 0.004 |
| Residence | | | | |
| Rural | 4 (5.4) | 1 (2.8) | ref | |
| Urban | 70 (94.6) | 35 (97.2) | 0.50 (0.05–4.64) | 0.542 |
| Monthly income of the Family (Euro) | | | | |
| ≤560 | 59 (79.7) | 25 (69.4) | ref | |
| >561 | 15 (20.3) | 11 (30.6) | 0.58 (0.23–1.43) | 0.236 |

*Values from the logistic regression analysis

**HTA; Chronic cardiac disease, except hypertension; diabetes; cancer; COPD and asthma and kidney diseases.

**Table 2. Bivariate analysis of vaccination status of cases and controls.**

| Variable | Cases | Controls | OR (95% CI) | p value* |
|---|---|---|---|---|
| | (N = 74) | (N = 36) | | |
| | No (%) | No (%) | | |
| Primary COVID-19 vaccine series before hospitalization (any type) | 39 (52.7) | 28 (77.8) | 0.32 (0.13–0.79) | 0.014 |
| Type primary COVID-19 vaccine series before hospitalization | | | | |
| Pfizer-BioNTech | 3 (7.9) | 4 (14.3) | 0.54 (0.11–2.66) | 445 |
| Astra Zeneca | 1 (2.6) | 0 (0.0) | - | 1.000 |
| Sputnjik V | 6 (15.8) | 4 (14.3) | 1.07 (0.27–4.30) | 922 |
| Sinopharm BBIBP-CorV | 28 (37.8) | 20 (55.5) | ref | |
| Third dose of COVID-19 vaccine before hospitalization (any type) | 24 (32.4) | 21 (58.3) | 0.34 (0.15–0.78) | 0.011 |
| Type of third dose of COVID-19 vaccine before hospitalization | | | | |
| Pfizer-BioNTech | 8 (33.3) | 4 (19.0) | 3.00 (0.71–12.70) | 0.136 |
| Astra Zeneca | 2 (8.3) | 0 (0.0) | - | 1.000 |
| Sputnjik V | 4 (16.7) | 2 (9.5) | 3.00 (0.46–19.60) | 0.251 |
| Sinopharm BBIBP-CorV | 10 (41.7) | 15 (71.4) | ref | |
| First booster dose of COVID-19 vaccine before hospitalization (any type) | 7 (9.5) | 4 (11.1) | 0.54 (0.14–2.08) | 0.368 |
| Type of first booster dose of COVID-19 vaccine before hospitalization | | | | |
| Pfizer-BioNTech | 4 (5.5) | 1 (2.8) | 4.00 (0.27–60.33) | 0.317 |
| Sinopharm BBIBP-CorV | 3 (4.1) | 3 (8.3) | ref | |
| Influenza vaccine | | | | |
| Season 2020/2021 | 16 (21.6) | 4 (11.1) | 2.21 (0.68–7.17) | 0.188 |
| Season 2021/2022 | 14 (18.9) | 5 (13.9) | 1.45 (0.49–4.54) | 0.477 |
| Season 2022/2023 | 3 (4.1) | 2 (5.6) | 1.57 (0.24–10.37) | 0.639 |

*Values from the logistic regression analysis

having received primary COVID-19 vaccine series (OR = 0.28; 95% CI = 0.09–0.88), current smoking (OR = 8.64; 95% CI = 2.43–30.72), previous SARS CoV-2 infection (OR = 3.48; 95% CI = 1.50–8.11) number of days before seeking medical help (OR = 0.81; 95% CI = 0.64–1.02), (Fig 1)). Model of multivariate logistic regression analysis was statistically significant and describes 45% of the variation of the dependent variable.

WGS was performed for 36 samples from April to December 2022 that met the defined criteria. Complete SARS-CoV-2 genome sequences were obtained for 22 samples and deposited in GISAID under Accession IDs: EPI_ISL_17075642–17075662 and EPI_ISL_17222305. The presence of 9 Omicron sublineages was detected where the sublineage BA.5.2 (22B) was the most frequent (40.9%), followed by BA.5.1 (22B) sublineage (22.7%) (Fig 2).

## Discussion

We identified age as independent predictor of COVID-19. This is not surprising because many of co-morbidities and risk factors for developing severe diseases, such as chronic cardiac disease, diabetes, cancer, COPD, asthma and kidney disease, occur more frequently in older than in younger patients [18, 19]. In our study, variable related to three of four comorbidities was very close to statistical significance (p = 0.008). Moreover, immunocompromised patients, usually common among elderly, are more likely to develop severe disseminated forms of disease and adverse drug reactions. Delay in diagnosis and treatment among older age groups is also common, which could increase the risk of death [20, 21]. Simultaneous presence of the diseases as a predictor for COVID-19 among SARI is based on the fact that non-

**Table 3. Bivariate analysis of disease specific factors and physical impairments among cases and controls.**

| Variable | Cases | Controls | OR (95% CI) | p value* |
|---|---|---|---|---|
| | (N = 74) | (N = 36) | | |
| | No (%) | No (%) | | |
| Current illness symptoms | | | | |
| Headache | 34 (45.9) | 14 (38.9) | 1.33 (0.59–3.01) | 0.484 |
| Sore throat | 36 (48.6) | 18 (50.0) | 0.95 (0.43–2.10) | 0.894 |
| Runny nose | 27 (36.5) | 16 (44.4) | 0.72 (0.32–1.61) | 0.423 |
| Shortness of breath | 43 (58.1) | 13 (36.1) | 2.45 (1.08–5.58) | 0.032 |
| General weakness and/or fatigue | 56 (75.7) | 26 (72.2) | 1.20 (0.49–2.96) | 0.697 |
| Muscle pains/myalgia | 35 (47.3) | 14 (38.9) | 1.41 (0.63–3.17) | 0.406 |
| Loss of smell (anosmia) | 8 (10.8) | 0 (0.0) | - | 0.999 |
| Loss of taste (ageusia) | 6 (8.1) | 0 (0.0) | | 0.999 |
| Vomiting or nausea or loss of appetite (anorexia) | 19 (25.7) | 10 (27.8) | 0.90 (0.37–2.20) | 0.814 |
| Abdominal pain | 7 (9.5) | 5 (13.9) | 0.65 (0.19–2.20) | 0.487 |
| Diarrhea | 9 (12.2) | 3 (8.3) | 1,52 (0.38–6.01) | 0.548 |
| Heart palpitations | 8 (10.8) | 5 (13.9) | 0.75 (0.23–2.49) | 0.640 |
| Chest pain | 7 (9.5) | 2 (5.6) | 1,78 (0.35–9,02) | 0.488 |
| Dizziness | 6 (8.1) | 3 (8.3) | 0.97 (0.23–4.13) | 0.968 |
| Number of days before seeking medical help, median (range) | 2 (1–10) | 2.5 (1–18) | 0.85 (0.72–0.99) | 0.044 |
| Period from symptom onset till hospital admission, median (range) | 3.5 (1–20) | 8 (1–20) | 0.81 (0.72–0.91) | <0.001 |
| Total number of in-hospital days for the present hospital stay, median (range) | 10 (1–65) | 9.5 (3–40) | 1.02 (0.98–1.07) | 0.350 |
| Oxygen saturation (SpO2) on admission (mean ±SD) | 93.3± 5.7 | 89.7± 7.4 | 1.09 (1.02–1.16) | 0.013 |
| Diagnosis of pneumonia among the patients | 51 (68.9) | 29 (80.6) | 0.46 (0.17–1.26) | 0.130 |
| Clinical diagnosis of pneumonia | | | | |
| Chest x-ray | 48 (64.9) | 26 (72.2) | 0.71 (0.30–1.70) | 0.441 |
| Chest CT | 5 (6.8) | 5 (13.9) | 0.45 (0.12–1.67) | 0.231 |
| Pneumonia severity index, median (range) | 10 (4–14) | 10.5 (8–15) | 0.89 (0.61–1.29) | 0.525 |
| Coagulation disorders | 52 (71.2) | 32 (91.4) | 0.23 (0.06–0.84) | 0.026 |
| Lowest recorded oxygen saturation during the hospitalization (mean, SD) | 91.0 ±5.4 | 88.4 ±7.8 | 1.07 (0.99–1.14) | 0.006 |
| Receive oxygen during hospital stay? | 35 (47.3) | 21 (58.3) | 0.64 (0.29–1.43) | 0.278 |
| Oxygen application method | | | | |
| Binasal cannula | 22 (29.7) | 14 (38.9) | 0.67 (0.29–1.53) | 0.338 |
| Oxygen mask | 13 (17.6) | 3 (8.3) | 2.34 (0.62–8.82) | 0.208 |
| Non-invasive ventilation | 0 (0.0) | 0 (0.0) | - | - |
| High flow | 1 (1.4) | 0 (0.0) | - | 1.000 |
| Mechanical/invasive ventilation | 3 (4.1) | 4 (11.1) | 0.34 (0.07–1.60) | 0.171 |
| Previous laboratory- confirmed SARS CoV-2 infection | 66 (89.2) | 13 (36.1) | 14.60 (5.37–39.70) | <0.001 |
| Test used for diagnose of previous SARS CoV-2 infection | | | | |
| Rapid test | 9 (13.6) | 0 (0.0) | ref | 0.0.999 |
| PCR | 57 (86.4) | 13 (100.0) | - | |
| Treatment outcome | | | | |
| Discharged alive | 71 (95.9) | 35 (97.1) | 1.39 (0.14–13.92) | 0.777 |
| Died in hospital | 3 (4.1) | 1 (2.9) | ref | |

* Values from the logistic regression analysis

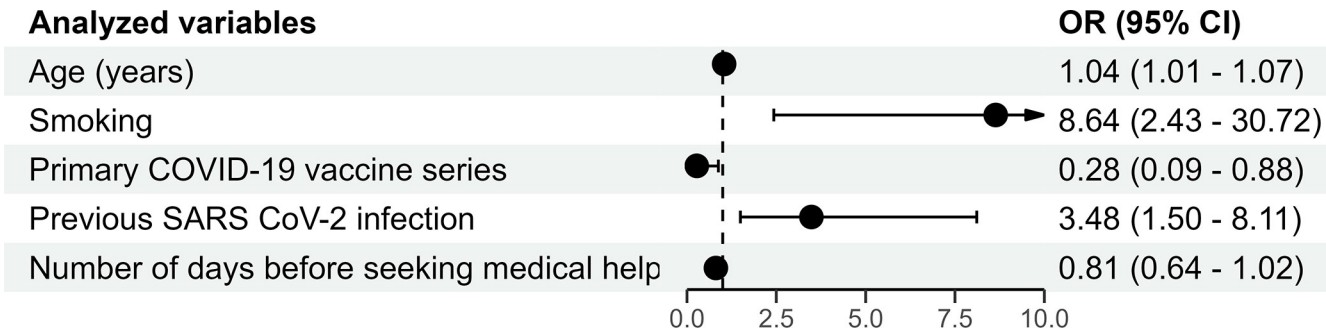

**Fig 1. Results of multivariable logistic regression analysis (dependent variable is SARS CoV-2 positive status).**

communicable diseases (NCD) multimorbidity accumulates the risks of the exposed host and increases its susceptibility for the infectious diseases' agents [22]. Namely, evidence has accumulated that inflammation contributes to the pathogenesis of the most common NCDs. Immune cells have been observed in vessels and kidneys of people affected in HTA. In addition, biomarkers of inflammation, including high sensitivity C-reactive protein, various cytokines, and products of the complement pathway are elevated in humans with HTA, diabetes, cancer, COPD and asthma [23, 24]. Dysregulation of host defence functions resulting in synergistic damage of the lungs, with interaction on immunologic and cellular levels leading to reduced function of the immune system are present in COPD and asthma. In addition, impaired muco-ciliary clearance, overall decreased mucosal defence function and long-term inhaled corticosteroids therapy increase potential possibility of SARSCoV-2 infection and development of COVID-19 [25].

We found that the primary series of COVID-19 vaccine administered before hospitalization was associated with non-COVID-19, in line with the results of many other studies [26, 27]. Although further stratification analysis could not be performed due to the low number of SARI patients, our study showed that vaccinated SARI patients are less likely to be affected by

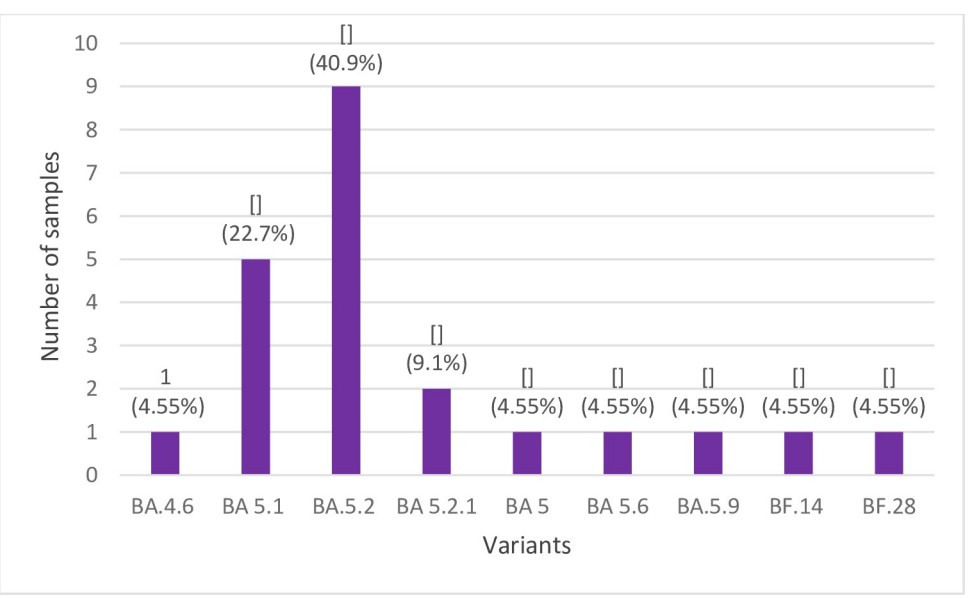

**Fig 2. The frequencies of identified Omicron variants.**

COVID-19 than non-vaccinated SARI patients. In general, our study results showed lower overall coverage of vaccination among study participants (60% for COVID-19 and 40% for influenza) than expected, given the fact that both vaccines are highly recommended for risk groups and patients aged 65+ and the majority of patients belong to that age group [28]. Future continuous nationwide efforts should be implemented in the promotion of COVID-19 vaccination in population and encouragement of rejection of misconceptions related to COVID-19 vaccines, particularly among high-risk groups.

Also, our study showed that previous laboratory confirmed SARS CoV-2 infection was predictor of re-infection with the same virus in line with the results of other studies. Systematic review and meta-analysis recently published in Lancet [29] showed that protection from pre-omicron variants was very high and remained so even after 40 weeks. However, it was substantially lower for the omicron BA.1 variant and declined more rapidly over time than protection against previous variants. This emphasizes the high immune escape features of this variant. Moreover, the dynamic of the protection is equivalent to the one achieved by primary series of mRNA vaccines and it has important influence on for guidance regarding the vaccine schedule, including booster doses.

In addition, number of days before seeking medical help is likely lower among COVID positive SARI patients than non-COVID. Late seeking of medical help is recognized as a common behavioural problem during the public health threats. At the beginning of the coronaviral disease pandemic, people avoid to visit health facilities in order not to be infected. As the pandemic spread and the overall knowledge and awareness raised, health seeking behaviour changed, particularly in cases of severe disease when people were looking for medical help much faster, as showed in the study of Kitazawa K. et al. [30].

Finally, current smoking was found to be significantly associated with COVID-19 among SARI patients. Robust evidence suggests that several different mechanisms might be responsible for the increased risk of respiratory tract infections in smokers. Smoking impairs the immune system and almost doubles the risk of latent tuberculosis infection and active disease. Precisely, smoking affects the macrophage and cytokine defence function and thus the ability to contain infection [31, 32]. Similarly, the risk for pneumococcal, legionella, and mycoplasma pneumonia infection is about 3–5 times higher in smokers. Users of tobacco and e-cigarettes have increased adherence of pneumococci and colonization, as a result of the upregulation of the pneumococcal receptor molecule (platelet activating receptor factor); smokers are also 5-times more likely to contract influenza than non-smokers [31, 33]. The largest study to date, from the UK, indicated associations with recent smoking behaviours and associations with lifelong predisposition to smoking and smoking heaviness support a causal effect of smoking on COVID-19 severity [34]. There is currently only limited information on COVID-19 in relation to other tobacco products (heated tobacco products, waterpipe, and cigars) and electronic nicotine delivery systems (e-cigarettes), although these products are thought to play an unfavourable role in COVID-19 severity [35].

This study contributed to increasing overall knowledge. Further follow up studies are needed to understand the further associations. However, this study did not indicate that sex, education level, BMI alcoholism and other clinical symptoms were significantly associated with COVID-19, as several other studies [20, 21, 36].

Until now, various Omicron sub lineages were circulating nationwide. WGS analysis of case samples from April to December 2022 revealed the presence of 9 Omicron sub lineages where B5.2 and B.5.1 were the most frequent. This is consistent with the data about frequencies of Omicron sub lineages from Serbia in GISIAD and Nextstrain database [37]. Epidemiological, molecular, and *in vitro* studies have demonstrated higher transmission rate of BA4 and BA5 sub lineages as compared to BA1 and BA2, where BA5 have had a higher fitness capacity

(38, 39). These data were supported with the high frequency rate of BA5 sub lineage globally. Several studies assessed risk factors for hospitalization and severity, comparing BA.1, BA2, BA.4 and BA.5 sub lineages. The results of these studies have shown that severe hospitalization (admission to intensive care or mechanical ventilation or oral/intravenous steroid prescription or mortality of BA.4/BA.5 was similar to the BA.1/BA2 sub lineages [38, 39].

The major strength of this study was inclusion of all SARI patients living in all parts of Serbia and selecting the cases and controls based on the result of molecular technique.

Several limitations of this study should be considered. Analysis of the effectiveness of different doses of different COVID-19 vaccines as a function of time before hospitalization, could not be investigated since low number of cases and controls. Information about influenza vaccination was collected by self-reporting and therefore, recall biases were unavoidable and potentially affected the results. Some information, such as income and use of alcohol, might be inaccurate as well due to self-reporting.

Despite these limitations, the risk factors associated with SARS CoV-2 among SARI patients provided important baseline information for planning of future interventions related to SARI prevention and control. To the best of our knowledge, the present case-control study is the first investigation of risk factors for COVID-19 among SARI patients in Serbia.

## Conclusions

In Serbia during a period of Omicron circulation, we found that older age, unvaccinated hospitalized SARI patients, previously infected with SARS CoV-2 virus and those who smoked were more likely to be SARS-CoV-2-positive; these patient populations should be prioritized for COVID vaccination.

## Supporting information

**S1 Data.**
(XLSX)

## Acknowledgments

The authors gratefully acknowledge the help of World Health Organization (WHO) colleagues Mark Katz, Iris Finci, Oksana Artemchuk, Amelia Casper, Katja Silling, Tobias Homan, clinicians Goran Stevanovic, Mihailo Stjepanovic, Milija Bjelicic, Igor Vujovic, Jelena Jankovic, Slobodan Belic, Boris Jegorovic and Teodora Cucanic, microbiologists Snezana Jovanovic, Ivana Pesic Pavlovic, Marko Jankovic, bioinformatician Ognjen Milicevic and SARI patients for their willingness to participate in the study.

## Author Contributions

**Conceptualization:** Maja Stosic.

**Data curation:** Dragana Plavsa, Marko Veljkovic, Aleksandra Knezevic.

**Formal analysis:** Maja Stosic, Dragan Babic.

**Investigation:** Dragana Plavsa, Marko Veljkovic, Aleksandra Knezevic, Tatjana Adzic-Vukicevic.

**Methodology:** Maja Stosic.

**Project administration:** Maja Stosic, Verica Jovanovic.

**Software:** Dragan Babic.

**Supervision:** Verica Jovanovic.

**Validation:** Dragana Plavsa.

**Visualization:** Aleksandra Knezevic.

**Writing – original draft:** Maja Stosic, Aleksandra Knezevic.

**Writing – review & editing:** Dragana Plavsa, Verica Jovanovic, Marko Veljkovic, Vladan Saponjic, Dragana Dimitrijevic, Miljan Rancic, Marija Milic, Tatjana Adzic-Vukicevic.

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
