## [Decision Letter · Decision Letter 0]

18 Dec 2023

PONE-D-23-39223Factors associated with COVID-19 among hospitalized patients with severe acute respiratory infections in Serbia, 2022-2023: a case-control studyPLOS ONE

Dear Dr. Stosic,

Thank you for submitting your manuscript to PLOS ONE. After careful consideration, we feel that it has merit but does not fully meet PLOS ONE’s publication criteria as it currently stands. Therefore, we invite you to submit a revised version of the manuscript that addresses the points raised during the review process.

We look forward to receiving your revised manuscript.

Kind regards,

Morteza Arab-Zozani, Ph. D.

Academic Editor

PLOS ONE

Journal Requirements:

2. In the online submission form, you indicated that "The data underlying the results presented in the study are available from the first author on a reasonable request".

Reviewers' comments:

Reviewer's Responses to Questions

**Comments to the Author**

1. Is the manuscript technically sound, and do the data support the conclusions?

Reviewer #1: Yes

Reviewer #2: No

2. Has the statistical analysis been performed appropriately and rigorously? 

Reviewer #1: Yes

Reviewer #2: No

3. Have the authors made all data underlying the findings in their manuscript fully available?

Reviewer #1: Yes

Reviewer #2: Yes

4. Is the manuscript presented in an intelligible fashion and written in standard English?

Reviewer #1: Yes

Reviewer #2: Yes

5. Review Comments to the Author

Reviewer #1: The authors have discussed a topic that will contribute to the scientific literature. Although Covid-19 has lost its former fame, it still has unsolved parts and this is why the study is important. In addition, the authors have made extremely statistically efficient analyzes and combined them with efficient tables. I believe that the study will contribute to the scientific literature.

Reviewer #2: 1. In introduction paragraph 2, reference 6 mentioned about SARI surveillance and not about COVID vaccine effectiveness.

2. In paragraph 4, its mentioned as data was collected from to SARI sentinel sites in Serbia, but the data as collected from two hospitals.

3. In inclusion and exclusion criteria paragraph, its detailed as those who had a history of hospitalization within the 14 days of current admission were excluded—why?

4. Sample size calculation is not mentioned. Number of cases were 74 which is less for a common disease like COVID-19 or SARI.

5. For a case control study, cases to controls ratio should be a minimum of 1:1 to have adequate power. Here controls are less than the cases.

6. Study looks like a test negative case control study, but it is not mentioned in the article.

7. Test negative case control study can be used for assessing vaccine effectiveness, but not usually used for finding the risk factors of a disease.

8. Different types of statistical tests are used in the same table, but it is not mentioned in the foot note with labelling.

9. In results paragraph 2, it is mentioned that cases were significantly more in those using tobacco products, but, in table 1, it is mentioned as history of current smoking and not as usage of tobacco products. Is it used synonymously?

10. In results paragraph 3, the percentage of SARI patients who received a third dose of COVID vaccine was similar between cases and controls. However, in table 2 it is given as 32.4% vs. 58.3%.

11. In results paragraph 4, aren’t the controls took a median of one day more to seek medical help and not cases?

12. In table 4, the variable of primary COVID-19 vaccine series before hospitalization is not significant as the confidence interval includes unity.

13. In discussion paragraph 2, SARI patients were 64% less likely to be affected by COVID-19; this data was not mentioned in the results.

14. In discussion paragraph 3, seeking medical help is 42% lower among COVID-19 SARI patients than non COVID-19—not mentioned in results.

15. In discussion, inclusion of patients living in all parts of Serbia is described as the strength of the study, which is not well founded.

16. This study was not able to identify any new risk factors of COVID-19.

17. Abbreviations like ULRA, MLRA and MVRA are not universally used and better be avoided.

6. PLOS authors have the option to publish the peer review history of their article (what does this mean?). If published, this will include your full peer review and any attached files.

Reviewer #1: **Yes: **Muhammed Emin DÜZ

Reviewer #2: **Yes: **Dr. Chitra Tomy

---

## [Author Response · Author response to Decision Letter 0]

5 Feb 2024

Corresponding Author:

Maja Stosic, MD, PhD, Assistant Professor

Institute of Public Health of Serbia “Dr Milan Jovanovic Batut”, Belgrade, Serbia

Tel/Fax: +381641278571

E-mail: maja_stosic@batut.org.rs

Belgrade, February 5, 2024

Dear Morteza Arab-Zozani, Ph. D. Academic Editor, PLOS ONE

We are pleased to submit our revised manuscript “Factors associated with COVID-19 among hospitalized patients with severe acute respiratory infections in Serbia, 2022-2023: a case-control study” by Stosic M. et al. for publication in the respectful journal such as PLOS ONE.

We are grateful for the comments of the reviewer and attentive review to improve our manuscript. We have replied point-by-point to all issues raised by the Reviewer.

Thank you for opportunity to revise our manuscript and we hope it will be suitable for publication in the PLOS ONE.

Sincerely,

Maja Stosic

Responses to Reviewers:

1. In introduction paragraph 2, reference 6 mentions about SARI surveillance and not about COVID vaccine effectiveness. 

ANSWER 1: Thank you for the comment. As recommended, we reordered the references within the Introduction section as well within the Reference section to ensure traceability.

2. In paragraph 4, it’s mentioned as data was collected from to SARI sentinel sites in Serbia, but the data was collected from two hospitals. 

ANSWER 2: Thank you for the comment. Two hospitals where the study was performed are part of the national SARI sentinel surveillance system in Serbia. Overall, six hospitals, (four hospitals treating adult patients and two treating child and adolescent patients) represent the national sentinel SARI surveillance network in Serbia.

During the COVID-19 pandemic, based on the clinical protocol, patients with the most severe clinical SARI presentations from all over the country were referred and treated in these two hospitals included in the study. In addition, third sentinel site was the "Batajnica" Hospital, dedicated for treatment of the most severe COVID-19 patients during this period. 

We provided clarification within the Introduction section as follows: Two hospitals where the study was performed are part of the national SARI sentinel surveillance system in Serbia. Overall, six hospitals, (four treating adult patients and two child and adolescent ones) represent the national sentinel SARI surveillance network in Serbia. During the COVID-19 pandemic, based on the clinical protocol, patients with the most severe clinical SARI presentations from all over the country were referred and treated in these two hospitals included in the study. In addition, third sentinel site was the "Batajnica" Hospital, dedicated for treatment of the most severe COVID-19 patients during study period.

3. In inclusion and exclusion criteria paragraph, its detailed as those who had a history of hospitalization within the 14 days of current admission were excluded—why?

ANSWER 3: We appreciate the comment. We excluded this sentence from the exclusion criteria since it was a mistake (typo). 

4. Sample size calculation was not mentioned. Number of cases were 74 which is less for a common disease like COVID-19 or SARI. 

ANSWER 4: Thank you for the comment. We included all eligible consecutive patients fulfilling the SARI case definition, we accessed during the study period. Due to extreme workloads, it was only feasible to switch from exhaustive to systematic sampling (e.g. inclusion of patients only once a week, on certain days). Based on the national surveillance data, during the COVID-19 pandemic, frequency of severe acute respiratory infections was much higher among SARS CoV-2 positive patients then in SARS CoV-2 negative patients, more than five times and therefore the number of cases is higher than the number of controls.

We provided this clarification within the Methods section-subsection Sample and Procedure. 

5. For a case control study, cases to controls ratio should be a minimum of 1:1 to have adequate power. Here controls are less than the cases.

ANSWER 5: Thank you for the comment. Based on the national surveillance data, during the COVID-19 pandemic, frequency of severe acute respiratory infections was much higher among SARS CoV-2 positive patients then in SARS CoV-2 negative patients, more than five times. Due to extreme workloads, it was not feasible to reach more patients. We found certain statistically significant differences among cases and controls even in a smaller sample, indicating that it is realistic that they would certainly be proven in a larger sample.

We provided clarification within the Methods section-subsection Sample and Procedure and within the ANSWER 4. 

6. Study looks like a test negative case control study, but it is not mentioned in the article. 

ANSWER 6: We appreciate the comment. We added this clarification throughout the text.

7. Test negative case control study can be used for assessing vaccine effectiveness, but not usually used for finding the risk factors of a disease.

ANSWER 7: We appreciate the comment. Although not frequently used in epidemiology these studies can give valid causal estimates of odds ratios and are used for risk factors analysis as per literature. 

Review article of Vandenbroucke JP. and Pearce N. (Vandenbroucke JP, Pearce N. Test-Negative Designs: Differences and Commonalities with Other Case-Control Studies with "Other Patient" Controls. Epidemiology. 2019 Nov;30(6):838-844), described the test-negative design as belonging to a family of similar designs where cases and controls are selected from patients from the same or similar healthcare facilities. It represents a subtype of this approach, where controls are patients with similar clinical signs and symptoms who have tested negative for the “case disease” in a further lab-based or imaging procedure. These studies can give valid causal estimates of odds ratios. Furthermore, valid population odds ratios can be estimated with controls that are not sampled from the source population, for example, with other patient controls, such as hospital controls. The views echoed by Westreich et al. and with several reviews that have upheld the validity of the test-negative design in risk factor analysis, among others by simulation studies and probability modeling. 

One of the examples: Ghanei M, Keyvani H, Haghdoost A, Abolghasemi H, Janbabaei G, Reza Jamshidi H, Hosein Ghazale A, Hassan Saadat S, Gholami Fesharaki M, Raei M. The risk factors and related hospitalizations for cases with positive and negative COVID-19 tests: A case-control study. Int Immunopharmacol. 2021 Sep; 98:107894. 

8. Different types of statistical tests are used in the same table, but it is not mentioned in the foot note with labelling.

ANSWER 8: Thank you for the comment. The p values were taken from the logistic regression analysis. Clarification was added in the tables. In accordance with the reviewers' requests related to statistical analysis and values, we consulted an additional statistician and performed complete statistics as requested by the reviewers. Changes and clarifications have been made to the method section – subsection statistical analysis and results section and the whole database is attached with the revision of the manuscript.

9. In results paragraph 2, it is mentioned that cases were significantly more in those using tobacco products, but, in table 1, it is mentioned as history of current smoking and not as usage of tobacco products. Is it used synonymously?

ANSWER 9. Thank you for the comment. Yes. We performed corrections in the text of the Results section to correspond exactly to the formulations provided in the tables. 

10. In results paragraph 3, the percentage of SARI patients who received a third dose of COVID vaccine was similar between cases and controls. However, in table 2 it is given as 32.4% vs. 58.3%.

ANSWER 10. Thank you for the comment. We performed certain corrections in the text of the Results section to correspond to the data provided in the tables since there were a few mistakes in the text. Kindly note, that the different terms are: primary series, third dose and booster dose. According to national COVID-19 vaccination guidelines, third dose was given as additional dose to the patients in which an adequate immune response was not achieved after the primary series (immunocompromised patients), while booster dose was given after at least three months from the primary series. 

Corresponding text is corrected as follows: In bivariate analysis, fewer cases had received primary vaccine series of any type and any type of third dose compared to controls; however, the percentage of SARI patients who had received a booster dose was almost similar between cases and controls. The majority of study participants (37.8% among cases and 55.5% among controls) received Sinopharm BBIBP-CorV as the primary series. (Table 2).

11. In results paragraph 4, aren’t the controls took a median of one day more to seek medical help and not cases?

ANSWER 11. Thank you for the comment. We corrected the mistake and corrected results. Now it is stated as follows: Controls took a median of one-half day more to seek medical help compared to cases (p = 0.044).

12. In table 4, the variable of primary COVID-19 vaccine series before hospitalization is not significant as the confidence interval includes unity.

ANSWER 12. We appreciate the comment. After we performed statistical analysis again, it was corrected. Now the value does not include unity. 

13. In discussion paragraph 2, SARI patients were 64% less likely to be affected by COVID-19; this data was not mentioned in the results.

ANSWER 13. Thank you for the comment. We corrected the mistake and excluded percentage in the statement. Now it is stated as follows: Although further stratification analysis could not be performed due to the low number of SARI patients, our study showed that vaccinated SARI patients are less likely to be affected by COVID-19 than non-vaccinated SARI patients.

14. In discussion paragraph 3, seeking medical help is 42% lower among COVID-19 SARI patients than non-COVID-19—not mentioned in results.

ANSWER 14: We appreciate the comment. We excluded the percentage in a formulation, since it was written by mistake as in the previous comment. The statement in a Discussion section is now as follows: In addition, number of days before seeking medical help is likely lower among COVID positive SARI patients than non-COVID.

15. In discussion, inclusion of patients living in all parts of Serbia is described as the strength of the study, which is not well founded.

ANSWER 15: Thank you for the comment. This statement is actually true. As explained under the comment number 2, during the COVID-19 pandemic, based on the clinical protocol, patients with the most severe clinical SARI presentations from all over the country were referred and treated in these two hospitals included in the study. In addition, third sentinel site, „Batajnica" Hospital, was dedicated for treatment of the most severe COVID-19 patients during this period, where patients from all over the country were referred for treatment. 

16. This study was not able to identify any new risk factors of COVID-19.

ANSWER 16: 

“Scientists have known for centuries that a single study will not resolve a major issue. Indeed, a small sample study will not even resolve a minor issue. Thus, the foundation of science is the cumulation of knowledge from the results of many studies.” (Hunter et al. 1982, p. 10).

From the beginning of COVID-19 pandemic, 409,314 different manuscripts have been published in Pub Med related to COVID-19 till the end of January 2024. The number of manuscripts related to risk factors for severe forms of COVID-19 in adults is big but lower, and there are no studies related to this topic from the context of Balkan countries so far. In addition, the aim of the study was not to recognize new risk factors for COVID-19 but to look at certain factors related to the context. Therefore, the importance of studies like ours is in contribution to increasing overall amount of knowledge about the risk factors for COVID-19 among severe acute respiratory infections, and the conclusions will be of importance in making trajectories of future public health efforts in COVID-19 prevention and control in our country and our region. 

17. Abbreviations like ULRA, MLRA and MVRA are not universally used and better be avoided.

ANSWER 17: Thank you for the comment. As suggested, we excluded the abbreviations throughout the text.

---

## [Editor Report · Decision Letter 1]

7 Feb 2024

Factors associated with COVID-19 among hospitalized patients with severe acute respiratory infections in Serbia, 2022-2023: a test negative case-control study

PONE-D-23-39223R1

Dear Dr. Stosic,

We’re pleased to inform you that your manuscript has been judged scientifically suitable for publication and will be formally accepted for publication once it meets all outstanding technical requirements.

Kind regards,

Morteza Arab-Zozani, Ph. D.

Academic Editor

PLOS ONE
---

## [Editor Report · Acceptance letter]

7 Mar 2024

PONE-D-23-39223R1 

PLOS ONE

Dear Dr. Stosic, 

I'm pleased to inform you that your manuscript has been deemed suitable for publication in PLOS ONE. Congratulations! Your manuscript is now being handed over to our production team.

Kind regards, 

on behalf of

Dr. Morteza Arab-Zozani 

Academic Editor

PLOS ONE